# Clinical and Oncological Outcomes Following Percutaneous Cryoablation vs. Partial Nephrectomy for Clinical T1 Renal Tumours: Systematic Review and Meta-Analysis

**DOI:** 10.3390/cancers16061175

**Published:** 2024-03-17

**Authors:** Łukasz Nowak, Dawid Janczak, Jan Łaszkiewicz, Maciej Guziński, Francesco Del Giudice, Anas Tresh, Benjamin I. Chung, Joanna Chorbińska, Wojciech Tomczak, Bartosz Małkiewicz, Tomasz Szydełko, Wojciech Krajewski

**Affiliations:** 1Department of Minimally Invasive and Robotic Urology, University Center of Excellence in Urology, Wrocław Medical University, 50-556 Wrocław, Poland; 2University Center of Excellence in Urology, Wrocław Medical University, 50-556 Wrocław, Poland; 3Department of General, Interventional and Neuroradiology, Wrocław Medical University, 50-367 Wrocław, Poland; 4Department of Maternal Infant and Urologic Sciences, “Sapienza” University of Rome, Policlinico Umberto I Hospital, 00161 Rome, Italy; 5Department of Urology, Stanford University School of Medicine, Stanford, CA 94305, USA

**Keywords:** renal cell carcinoma, percutaneous cryoablation, partial nephrectomy, complications, renal function, oncological outcomes

## Abstract

**Simple Summary:**

Percutaneous cryoablation (PCA) is a minimally invasive procedure that should be considered in comorbid patients with stage T1 renal tumours who are suboptimal candidates for partial nephrectomy (PN). However, there is a scarcity of scientific data regarding the efficacy of PCA. The aim of this meta-analysis was to compare PCA and PN in terms of complications, renal function and survival outcomes. According to this analysis, PCA is associated with fewer complications than PN. Moreover, in tumours up to 4 cm, it provides the same time without local recurrence. Therefore, PCA should be proposed to patients with cT1 renal tumours who are not fit for PN but want to undergo a radical treatment.

**Abstract:**

Percutaneous cryoablation (PCA) can be an alternative to partial nephrectomy (PN) in selected patients with stage T1 renal tumours. Existing meta-analyses regarding ablative techniques compared both laparoscopic and PCA with PN. That is why we decided to perform a meta-analysis that focused solely on PCA. The aim of this study was to compare the complications and functional and oncological outcomes between PCA and PN. A systematic literature search was performed in January 2024. Data for dichotomous and continuous variables were expressed as pooled odds ratios (ORs) and mean differences (MDs), both with 95% confidence intervals (CIs). Effect measures for the local recurrence-free survival (LRFS), metastasis-free survival (MFS), cancer-specific survival (CSS) and overall survival (OS) were expressed as pooled hazard ratios with 95% CIs. Among 6487 patients included in the 14 selected papers, 1554 (23.9%) and 4924 (76.1%) underwent PCA and PN, respectively. Compared with the PN group, patients undergoing PCA had significantly lower overall and major postoperative complication rates. There was no difference in renal function between PCA and PN groups. When analysing collective data for cT1 renal carcinoma, PCA was associated with worse LRFS compared with PN. However, subgroup analysis revealed that in the case of PCA, LRFS was not decreased in patients with cT1a tumours. Moreover, patients undergoing robotic-assisted PN had improved LRFS compared with those undergoing PCA. No significant differences were observed between PCA and PN in terms of MFS and CSS. Finally, PCA was associated with worse OS than PN in both collective and subgroup analyses. In conclusion, PCA is associated with favourable postoperative complication rates relative to PN. Regarding LRFS, PCA is not worse than PN in cT1a tumours but has a substantially relevant disadvantage in cT1b tumours. Also, RAPN might be the only surgical modality that provides better LRFS than PCA. In cT1 tumours, PCA shows MFS and CSS comparable to PN. Lastly, PCA is associated with a shorter OS than PN.

## 1. Introduction

Renal cell carcinoma (RCC) constitutes 3% of all cancers [1]. Surgery is considered the only radical treatment in localised RCC, with partial nephrectomy (PN) being recommended by the European Association of Urology (EAU) in T1 tumours [1]. However, in selected patients with cT1 (especially cT1a) disease, focal therapy, such as percutaneous cryoablation (PCA), can be used as an alternative to PN. PCA is gaining popularity due to the low risk of complications and kidney function deterioration [1].

PCA uses extreme cold, which destroys tumour cells both directly and by the initiation of apoptosis [2]. What is more, the procedure releases native tumour antigens, which may enhance the post-ablative immunologic response and cause an abscopal effect [3].

Over the past several years, significant advancements in cryoablation technology have broadened the treatment landscape for renal tumours. Technical developments in cryoablation devices, such as cryoprobes refinement, real-time visualisation, implementation of augmented reality systems or improved energy delivery mechanisms, have played a pivotal role in enhancing treatment efficacy, precision and patient outcomes [4,5]. The ongoing innovation and refinement of cryoablation technology continue to expand the therapeutic options available to patients with renal tumours, paving the way for improved treatment outcomes and enhanced quality of life. Therefore, there is currently a high level of research interest in comparing percutaneous cryoablation with operative approaches for the treatment of renal tumours.

To date, existing meta-analyses either compared PN with ablative therapy in general or combined data for both laparoscopic and percutaneous approaches for cryoablation. Hence, to avoid significant procedural heterogeneity, the aim of this systematic review and meta-analysis was to compare postoperative complication rates, as well as functional and oncological outcomes between PCA and PN in patients with cT1 renal masses.

## 2. Materials and Methods

The present systematic review and meta-analysis was performed in accordance with the Preferred Reporting Items for Systematic Reviews and Meta-Analyses (PRISMA) statement [6]. The study protocol was registered a priori on the International Prospective Register of Systematic Reviews (PROSPERO) (assigned registration number: CRD42023475443).

### 2.1. Search Strategy

Three review authors (ŁN, JŁ and WK) independently conducted the comprehensive literature search on the three electronic databases (PubMed, Embase and Cochrane Library) using the following search string: (“cryoablation” OR “percutaneous cryoablation” OR “CA” OR “PCA”) AND (“partial nephrectomy” OR “nephron-sparing surgery” OR “PN” OR “NSS”) AND (“kidney” OR “renal”) AND (“cancer” OR “neoplasm” OR “tumour” OR “tumor” OR “mass”). The last search was performed on 31 January 2024. There were no limitations imposed on time, geography or language as long as an abstract in English was present in the analysed study. A cross-reference search was also performed on articles selected for full-text review.

### 2.2. Inclusion and Exclusion Criteria

The eligibility of studies was evaluated using the PICOS (population, intervention, comparison, outcome, study design) approach. The criteria for inclusion in the present systematic review and meta-analysis were as follows:(P)opulation: patients with a cT1 renal tumour;(I)ntervention: patients who underwent PCA;(C)omparison: patients who underwent PN;(O)utcome: Postoperative complications (overall and Clavien-Dindo ≥ 3), renal function (% eGFR preservation and/or mean change in eGFR after surgery), oncological outcomes (local recurrence-free survival (LRFS), metastasis-free survival (MFS), cancer-specific survival (CSS) and overall survival (OS). LRFS was defined as the time from PCA or PN to any local recurrence. MFS was defined as the time from PCA or PN to the first evidence of distant metastatic disease. CSS was defined as the time from PCA or PN to documented death from RCC. OS was defined as the time from PCA or PN to documented death for any reason.For clinical and functional outcomes (postoperative complications and renal function), analyses were performed using data of patients with benign, malignant or unknown histological status. Only data for histologically confirmed RCC patients were included in the analyses of oncological outcomes;(S)tudy design: randomised controlled trials (RCTs), non-randomised observational cohorts and population-based cohorts.

### 2.3. Data Extraction

Two review authors (ŁN and JŁ) separately extracted a predefined set of data from included studies using standard data extraction templates. When multiple reports of the same cohort were available, the one with the most complete data aggregated and with the longest follow-up duration was selected (in the case of the same analysed outcomes). For each selected study, the following items were initially retrieved: study-related data (first author, publication year, journal, country, design and duration of the study, number of patients), clinicopathological data (age, gender, PN approach, Charlson Comorbidity Index (CCI) score, proportion of patients with solitary kidney, preoperative estimated glomerular filtration rate (eGFR), proportion of patients who underwent tumour biopsy before PCA, clinical tumour stage, tumour size, RENAL nephrometry score, proportion of patients with malignant histology, tumour grade, proportion of patients with residual unablated tumour after PCA, proportion of patients with positive margin after PN and follow-up duration). Subsequently, data regarding outcomes of interest were extracted, including complication rates, change in eGFR after surgery and oncological outcomes (hazard ratios (HRs) and 95% confidence intervals (CIs) associated with LRFS, MFS, CSS and OS). Throughout the whole data extraction process, discrepancies were resolved by discussion with a senior co-investigator (WK).

### 2.4. Quality Assessment and Risk of Bias

The risk of bias (RoB) evaluation for each study was assessed according to the Cochrane Handbook for Systematic Reviews of Interventions [7]. Initially, we assessed the potential presence of selection bias (random sequence generation and allocation concealment), performance bias, detection bias, attrition bias, reporting bias and other sources of bias. Due to the inclusion of non-randomised comparative studies, RoB was additionally determined by examining the implementation of matching techniques. The RoB of each study was assessed independently by two authors (ŁN and JŁ). Disagreements were resolved by consensus or consultation with the senior author (WK).

### 2.5. Statistical Analysis

All statistical analyses were performed using Review Manager 5.4 (The NordicCochrane Center, The Cochrane Collaboration, Copenhagen, Denmark), Statistica 13.3 (StatSoft Inc., Tulsa, OK, USA) and Stata 16.0 software (STATA Corporation, College Station, TX, USA).

Data for dichotomous variables (complication rates) were combined using the Mantel–Haenszel method and expressed as pooled odds ratios (ORs) with 95% CIs. In the case of continuous variables (% eGFR preservation and mean change in eGFR after surgery), results were calculated using the inverse variance (IV) method and presented as pooled mean differences (MDs) with a 95% CIs. If an article contained data reported as median and interquartile range (IQR), mean and standard deviations (SDs) were calculated by methods described by Luo et al. [8] and Wan et al. [9]. Effect measures for RFS, MFS, CSS and OS were HRs and 95% CIs extracted from selected studies (preferably from PSM or IPTW adjusted or multivariable analyses, if reported). For papers that did not directly provide HRs and 95% CIs, data from the presented Kaplan–Meier curves were extracted and calculated using methods described by Tierney et al. [10]. The statistical significance of the pooled HRs was evaluated by the Z test. Evaluation of the presence of heterogeneity was performed using Cochran’s Q-test and the Higgins I^2^ test. Significant heterogeneity was indicated by either a ratio of >50% in I^2^ statistics and a *p*-value < 0.05 in Cochran’s Q-test, which led to the use of the random effect (RE) model. Otherwise, the fixed effect (FE) model was used in analyses. Following assessment of primary analyses, prespecified exploratory subgroup analyses, including stratification by clinical stage and PN technique, were performed. Publication bias for each comparison was evaluated using either a visual assessment of funnel plots or results of the Egger’s tests. For all tests, a *p*-value ≤ 0.05 was considered a statistically significant difference.

## 3. Results

### 3.1. Search Results

A detailed flow diagram of the study selection process (with subsequent exclusions) is presented in Figure 1. During the initial systematic literature search, 3490 publications were identified. Following the removal of duplicates, a total of 2985 articles were available. After an extensive screening of the titles and abstracts, 2001 studies were excluded due to inappropriate study types, and 889 records were eliminated due to irrelevance to the present topic. Full-text reviews were performed for the remaining 95 articles. Ultimately, 81 papers were excluded (other than percutaneous cryoablation technique, single-arm studies, studies reporting no single outcome of interest or studies with overlapping cohorts), leaving 14 studies included in the present systematic review and meta-analysis [11,12,13,14,15,16,17,18,19,20,21,22,23,24].

### 3.2. Study Characteristics and Risk of Bias Assessment

The main summary characteristics of the included studies are shown in Table 1. Among a total of 6487 participants in the selected articles, 1554 (23.9%) underwent PCA, whereas 4924 (76.1%) had PN. The available records were published from 2017 to 2023. Seven out of fourteen studies [14,15,16,17,18,21,22] presented data for European populations, followed by four [11,19,23,24] and three [12,13,20] papers conducted in Japan and the United States, respectively. The vast majority of studies had retrospective designs [11,12,13,14,15,17,19,20,22,23,24]. In six articles [11,12,13,14,18,20], PN was performed using either an open or minimally invasive (laparoscopic or robotic) approach, while six [16,17,19,21,22,23] and two [15,24] trials included patients undergoing only robotic and laparoscopic PN, respectively. Detailed data on PCA systems and techniques are summarised in Appendix A.

The characteristics of tumour features and pathologic outcomes presented in selected articles are summarised in Table 2. The majority of publications [13,14,16,17,18,19,23,24] focused on individuals with either cT1a or cT1b renal tumours, yet the proportion of cT1b renal masses was relatively low in these investigations. Clinical data and outcomes for both cT1a and cT1b renal cell tumours were reported separately in 2 out of 14 articles [12,15], while 2 articles [11,22] analysed only patients with cT1b tumours and 1 article [21] exclusively analysed patients with cT1a tumours. A male predominance was observed in all studies. Generally, patients undergoing PCA were older and had a higher CCI score, along with a lower preoperative eGFR value. Data regarding the proportion of patients undergoing biopsy before PCA was reported in 11 studies [11,12,14,15,16,17,18,19,22,23,24]. Of these, eight manuscripts [14,15,16,17,18,19,22,23] described a 100% biopsy rate. The proportion of patients with histologically confirmed RCC varied from 48 to 100% and 72 to 100% in the PCA and PN groups, respectively. The rates of residual unablated tumours after PCA and positive surgical margins after PN were scarcely presented, ranging from 6.9 to 8.5% and from 1.3 to 5.6%, respectively. The duration of follow-up ranged from 12 to 75.6 months in the PCA group and from 18 to 112.8 months in the PN group. All selected studies, including prospective trials, were characterised by high RoB. Of the 11 retrospective papers, 7 utilised matching techniques [11,12,13,17,20,22,24].

### 3.3. Meta-Analysis Results

#### 3.3.1. Complications

Data on overall complication rates were extracted from eight articles [11,13,15,17,18,21,22,24]. When analysing collective data for both cT1a and cT1b tumours, PCA was associated with significantly lower complication rates compared with PN (RE model: OR: 0.63, 95% CI: 0.40–0.99, *p* = 0.05) (Figure 2A). The Cochrane’s Q (*p* = 0.04) and I^2^ (I^2^ = 51%) tests revealed significant heterogeneity between the included studies. Examination of the funnel plot (Appendix A) combined with analysis of the Egger’s test result did not demonstrate a significant publication bias.

Data on major complication rates (Clavien-Dindo ≥ 3) were extracted from 10 articles [11,13,15,17,18,19,21,22,23,24]. When analysing collective data for both cT1a and cT1b tumours, PCA was associated with significantly lower major complication rates compared with PN (FE model: OR: 0.39, 95% CI: 0.22–0.70, *p* = 0.002) (Figure 2B). The Cochrane’s Q (*p* = 0.82) and I^2^ (I^2^ = 0%) tests revealed no significant heterogeneity between the included studies. Examination of the funnel plot (Appendix A) combined with analysis of the Egger’s test result did not demonstrate a significant publication bias.

The detailed results of prespecified subgroup analyses are presented in Appendix A. No statistically significant differences in overall and major complication rates were found for homogenous cT1a and cT1b cohorts. Similar to the main analysis, data from studies reporting mixed cohorts in terms of clinical stage (cT1a and cT1b) indicated lower overall and major complication rates in the PCA group relative to the PN group.

#### 3.3.2. Renal Function

Data for renal function was extractable from nine studies [11,14,15,16,20,21,22,23,24], of which four [11,22,23,24] reported % eGFR preservation and five [14,15,16,20,21] reported mean (or median) change in eGFR following surgery. When analysing collective data for both cT1a and cT1b tumours, no significant difference in mean % eGFR preservation rates between PCA and PN groups was found (FE model: MD: −0.28, 95% CI: −2.58 to 2.03, *p* = 0.81, I^2^ = 28%) (Figure 3A). Also, the mean change in eGFR after surgery was comparable between PCA and PN groups (RE model: MD: 3.83, 95% CI: −1.24 to 8.93, *p* = 0.14, I^2^ = 97%) (Figure 3B). Examination of the funnel plot (Appendix A) combined with analysis of the Egger’s test result did not demonstrate a significant publication bias. Prespecified subgroup analyses could not be reliably performed due to the low number of studies in particular subgroups.

#### 3.3.3. Oncological Outcomes

Data for LRFS was extracted from 10 studies [11,12,13,14,15,17,19,22,23,24]. When analysing collective data for both cT1a and cT1b tumours, the pooled results indicated that compared with PN, PCA was associated with significantly worse LRFS (FE model: HR: 2.39, 95% CI: 1.67–3.41, *p* < 0.001) (Figure 4A). The Cochrane’s Q (*p* = 0.12) and I^2^ (I^2^ = 34%) tests revealed no significant heterogeneity between the included studies. Examination of the funnel plot (Appendix A) combined with analysis of the Egger’s test result did not demonstrate a significant publication bias.

The detailed results of prespecified subgroup analyses for LRFS are presented in Appendix A. For cT1a tumours, no statistically significant difference in LRFS was observed between PCA and PN groups (FE model: HR: 1.66, 95% CI: 0.63–4.39, *p* = 0.31). Similar to the main analysis, pooled data from studies reporting homogenous cT1b cohorts and mixed cohorts in terms of clinical stage (cT1a and cT1b) indicated worse LRFS in the PCA group relative to the PN group. Subgroup analysis of various PN approaches revealed that patients with cT1 renal masses undergoing RAPN had significantly improved LRFS compared with those undergoing PCA. However, pooled data from studies reporting homogenous LPN cohorts and mixed cohorts in terms of surgical approach (open PN and/or RAPN and/or LPN) indicated comparable LRFS between PCA and PN groups.

Data for MFS was extractable from four studies [11,12,13,15]. When analysing collective data for both cT1a and cT1b tumours, the pooled results indicated that compared with PN, PCA was not associated with significantly worse MFS (FE model: HR: 0.72, 95% CI: 0.43–1.19, *p* = 0.2) (Figure 4B). The Cochrane’s Q (*p* = 0.29) and I^2^ (I^2^ = 19%) tests revealed no significant heterogeneity between the included studies. Examination of the funnel plot (Appendix A) combined with analysis of the Egger’s test result did not demonstrate a significant publication bias. Similar to the main analysis, no statistically significant differences in MFS were found between PCA and PN in subgroup analyses of the homogenous cT1a and cT1b cohorts (Appendix A).

Data for CSS was extractable from four studies [11,12,13,15]. When analysing collective data for both cT1a and cT1b tumours, the pooled results indicated that compared with PN, PCA was not associated with significantly worse CSS (FE model: HR: 1.11, 95% CI: 0.60–2.07, *p* = 0.73) (Figure 4C). The Cochrane’s Q (*p* = 0.88) and I^2^ (I^2^ = 0%) tests revealed no significant heterogeneity between the included studies. Examination of the funnel plot (Appendix A) combined with analysis of the Egger’s test result did not demonstrate a significant publication bias. Similar to the main analysis, no statistically significant differences in CSS were found between PCA and PN in subgroup analyses of the homogenous cT1a and cT1b cohorts (Appendix A).

Data for OS was extractable from four studies [11,12,13,15]. When analysing collective data for both cT1a and cT1b tumours, the pooled results indicated that patients undergoing PCA had significantly worse OS compared with patients treated with PN (FE model: HR: 2.08, 95% CI: 1.67–2.60, *p* < 0.001) (Figure 4D). The Cochrane’s Q (*p* = 0.30) and I^2^ (I^2^ = 17%) tests revealed no significant heterogeneity between the included studies. Examination of the funnel plot (Appendix A) combined with analysis of the Egger’s test result did not demonstrate a significant publication bias. Similar to the main analysis, statistically significant differences in OS were found between PCA and PN in subgroup analyses of the homogenous cT1a and cT1b cohorts (Appendix A).

## 4. Discussion

According to the EAU guidelines, tumour ablation may be considered in comorbid patients with small renal masses [1]. Nonetheless, the strength of these recommendations is weak due to the scarcity of data regarding CA [1]. There is a lack of large high-quality trials that are necessary in order to precisely assess the clinical usefulness of CA. A meta-analysis by Deng et al. from 2019 proved that PCA, when compared with PN, is associated with poorer oncological outcomes but significantly lower risk of complications and better kidney function preservation [25]. Yet, this review included both laparoscopic and percutaneous cryoablations. The present systematic review and meta-analysis focused solely on the functional and oncological outcomes of PCA and PN.

In the available literature, PN is as effective as radical nephrectomy (RN) with regards to LRFS and CSS in localised renal masses [26,27]. A few studies suggested that PN may be associated with shorter OS [27,28]. On the other hand, a meta-analysis by Gu et al. suggested that PN can improve OS by 19% in comparison to RN [26]. Moreover, no differences in oncological outcomes (progression-free survival, cancer-specific mortality, overall mortality) were found specifically in stage T1b renal tumours [29]. Also, PN allows the preservation of functional kidney tissue, which lowers the risk of chronic kidney disease and its complications [29]. Considering these facts, PN should be a treatment of choice in patients with T1 RCC.

Nonetheless, in patients with T1 tumours who are unfit for PN procedure, even less toxic treatment, such as PCA, might be considered. The results of our systematic review suggest that PCA is a reasonable approach in localised RCC, especially in the cT1a stage. Our meta-analysis showed that LRFS in the cT1a subgroup did not differ between PCA and PN. However, LRFS was decreased in cT1b tumours in the PCA group. These findings are coherent with EAU guidelines that recommend not performing CA routinely in tumours larger than 4 cm [1].

Interestingly, when subgroup analysis based on surgical modalities was performed, patients who underwent RAPN had significantly longer LRFS compared with PCA for both cT1a and cT1b tumours. Other surgical methods of PN were not beneficial compared with PCA with regards to LRFS in cT1 tumours. Nevertheless, the studies specifically comparing the outcomes of RAPN, LPN and OPN found that the surgical approach did not influence LRFS [30,31,32].

What is worth mentioning is that in this study, we found no differences in CSS and MFS between the groups and worse OS in PCA patients. In the other meta-analyses comparing laparoscopic and percutaneous CA with PN, CA was associated with significantly worse RFS, MFS, PFS, OM and CSM [25,33,34]. These reviews, however, did not present subgroup analyses for cT1a/cT1b tumours, and some did not specify the stage of the included tumours.

PCA procedure is truly minimally invasive. During PCA, the needle, sized between 17 and 10 Gauge, is introduced into the tumour under CT and/or USG guidance. Then, the freezing process is initiated, forming an “ice ball” that can be observed in real time. This “live” control of the ice ball size increases the probability of obtaining safe surgical margins and conserving as much kidney tissue as possible [2]. Moreover, except for the introduction of needles, the process of cryoablation does not cause pain in patients. Therefore, in imperative clinical scenarios, PCA might be executed using only local lidocaine analgesia. As a result, PCA can be performed in patients who are not optimal candidates for general surgery, such as PN. Also, PN is associated with a significant risk of complications, both long-term and perioperative. According to a recent review, major complications affected 6.6–10% of PN, depending on the operative technique [35]. On the other hand, the reported incidence of major complications in PCA is 0–7.2% [36]. The risk factors of PCA complications are age, comorbidity, tumour size, upper pole location, number of cryoprobes, previous renal surgeries, RENAL score and (MC)2 score [36]. Nevertheless, it is crucial to always perform a biopsy before PCA in order to implement optimal disease management and reduce the risk of overtreatment and unnecessary follow-up [1].

In our meta-analysis, we showed that PCA had some clear advantages over PN, such as a significantly lower rate of overall and major postoperative complications. These results were similar in the analyses comparing laparoscopic CA and PN, even though laparoscopic CA is notably more invasive than PCA [33,34]. Therefore, we could assume that PCA is a safer procedure than PN. Moreover, other studies suggested that PCA generated lower costs than PN, with almost no differences in quality of life between the procedures [37,38,39].

The observed heterogeneity in analysed outcomes, encompassing both perioperative and oncological parameters, was generally low to moderate, with I2 values predominantly below 50%. This suggests that the combined results derived from the included studies can be interpreted reliably and potentially incorporated into clinical practice. The overall consistency in the direction and magnitude of effects across studies supports the robustness of our meta-analysis findings. To further explore potential sources of heterogeneity and understand any imbalances between results, we conducted detailed subgroup analyses. These analyses revealed that disparities in outcomes predominantly stemmed from the inclusion of tumours at different clinical stages (cT1a and/or cT1b) and variations in surgical approach. While our subgroup analyses provided valuable insights into potential sources of heterogeneity, it is essential to acknowledge certain limitations. The heterogeneity observed across different clinical stages and surgical approaches underscores the importance of conducting future studies and meta-analyses with more homogeneous patient populations. By focusing on uniform groups in terms of clinical stage and surgical approach, future research should provide more precise and reliable estimates of treatment effects.

This analysis has some limitations that need to be disclosed. The studies used in this meta-analysis were retrospective and heterogeneous. The PCA groups included in this review were relatively small. Most studies did not report all of the analysed outcomes. Due to possible selection bias, younger patients with fewer comorbidities could have been selected for PN, while elderly patients with more comorbidities could have undergone PCA. Therefore, all of the outcomes in the PCA group could have been inherently worse, regardless of the surgical method. A few studies did not report performing a biopsy before PCA, and the majority did not specify tumour grades. PCA and PN techniques varied across the included studies. Lastly, we could not obtain all the information on the technical aspects of cryoablation, such as the system used, needle size, freezing time etc.

## 5. Conclusions

Our present findings confirm that PCA is associated with favourable postoperative complication rates relative to PN. According to this analysis, PCA does not preserve kidney function better than PN. Regarding LRFS, PCA is not worse than PN in cT1a but has a substantially relevant disadvantage in cT1b renal masses. Also, RAPN might be the only surgical modality that provides better LRFS than PCA. In cT1 tumours, PCA shows MFS and CSS comparable to PN. Lastly, PCA is associated with a shorter OS than PN. Therefore, PCA is a viable treatment in comorbid patients with T1 renal tumours.

## Figures and Tables

**Figure 1 cancers-16-01175-f001:**
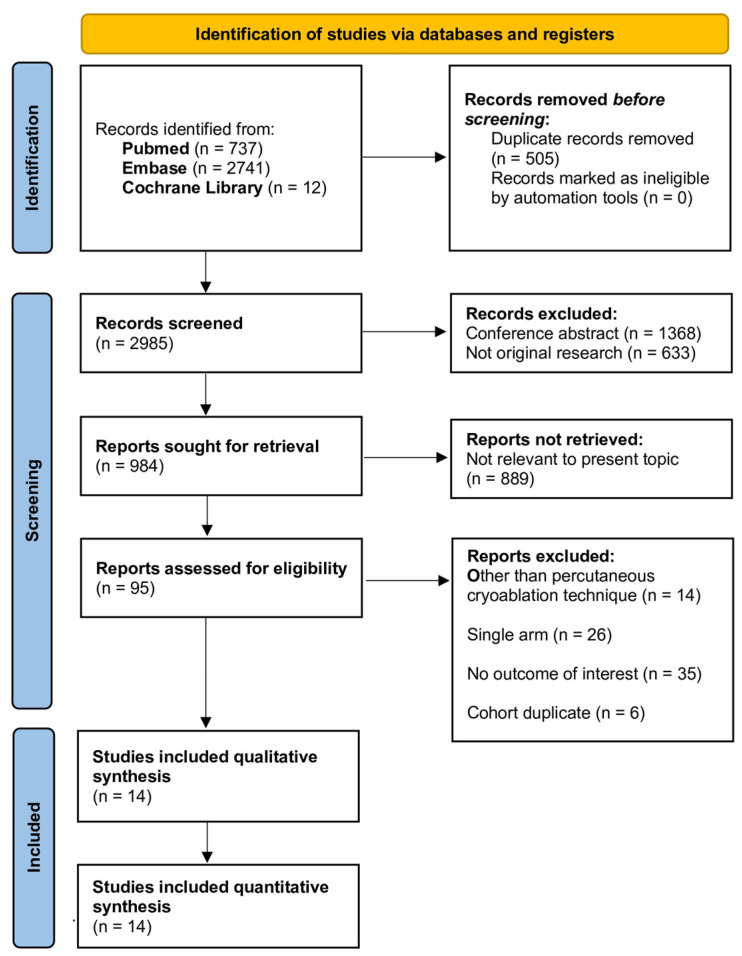
The Preferred Reporting Items for Systematic Reviews and Meta-analyses (PRISMA) flow chart detailing the article selection process.

**Figure 2 cancers-16-01175-f002:**
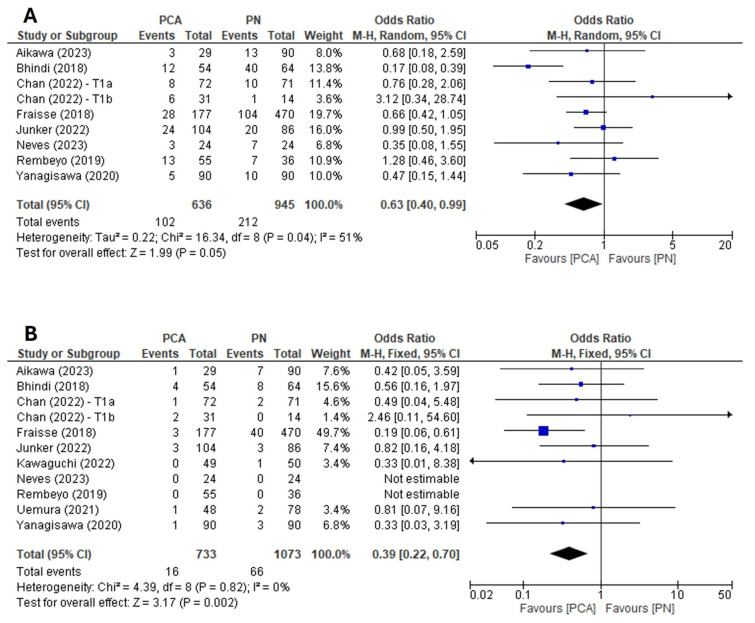
Forest plot of the odds ratio for (**A**) overall complication rates and (**B**) major complication rates [11,13,15,17,18,19,21,22,23,24].

**Figure 3 cancers-16-01175-f003:**
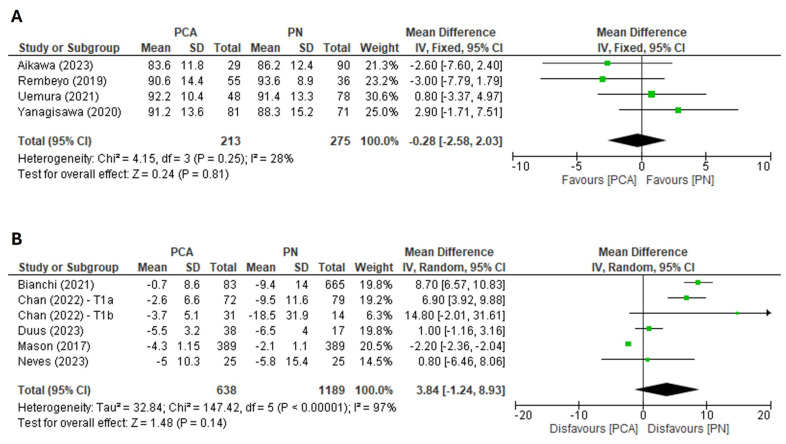
Forest plot of the mean difference for (**A**) % eGFR preservation and (**B**) change in eGFR following surgery [11,14,15,16,20,21,22,23,24].

**Figure 4 cancers-16-01175-f004:**
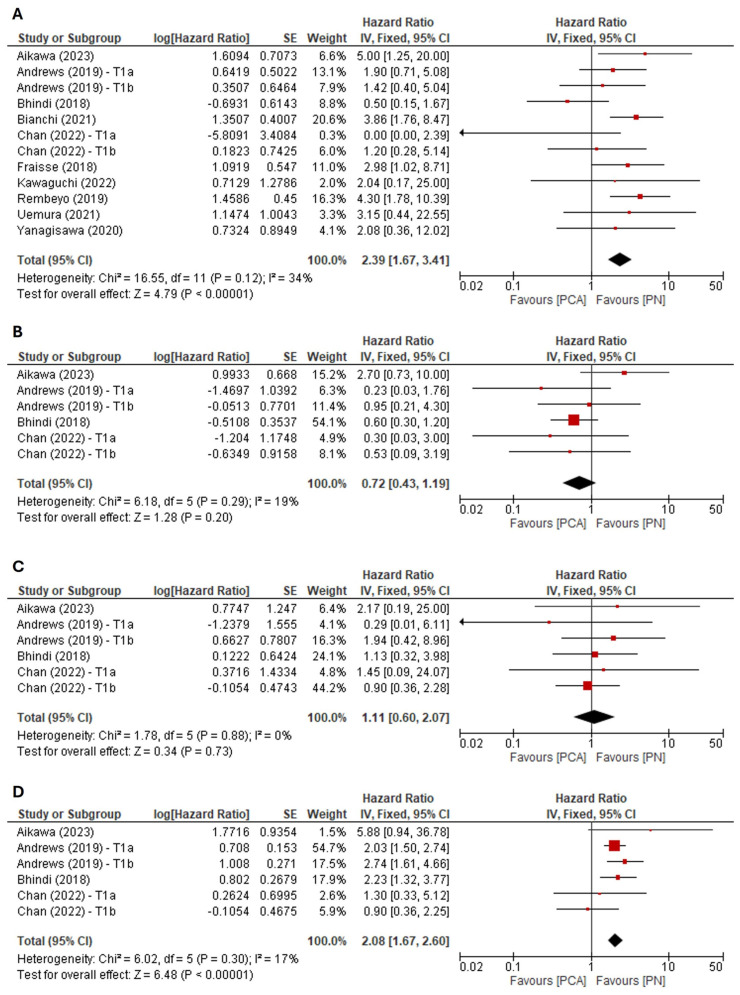
Forest plot of the hazard ratio for (**A**) local recurrence-free survival, (**B**) metastasis-free survival, (**C**) cancer-specific survival and (**D**) overall survival [11,12,13,14,15,17,19,22,23,24].

**Table 1 cancers-16-01175-t001:** Baseline characteristics of the included studies.

Author [Ref.]	Publication Year	Country	Study Design	Study Interval	PN Approach	No. of Patients PCA Group, *n*	No. of Patients PN Group, *n*	Outcomes of Interest
Aikawa et al. [9]	2023	Japan	Retrospective, multi-centre IPTW	2011–2021	OPN, LPN	29	90	1, 2, 3, 4, 5, 6
Andrews et al. [10]	2019	United States	Retrospective, single-centre PSM	2000–2011	OPN, LPN, RAPN	239	1429	3, 4, 5, 6
Bhindi et al. [11]	2018	United States	Retrospective, single-centre IPTW	2005–2015	OPN, LPN, RAPN	64	54	1, 2, 3, 4, 5, 6
Bianchi et al. [12]	2021	Italy	Retrospective, multi-centre	2007–2019	OPN, LPN, RAPN	83	665	2, 3, 4, 5
Chan et al. [13]	2022	United Kingdom	Retrospective, single-centre	2003–2016	LPN	103	93	1, 2, 3, 4, 5, 6
Duus et al. [14]	2023	Denmark	Prospective, single- centre	2019–2021	RAPN	38	18	2
Fraisse et al. [15]	2018	France	Retrospective, multi-centre PSM	2009–2016	RAPN	177	470	1, 3, 4, 5
Junker et al. [16]	2022	Denmark	Prospective, multi-centre	2019–2021	OPN, RAPN	101	86	1
Kawaguchi et al. [17]	2022	Japan	Retrospective, single-centre	2016–2021	RAPN	49	50	1, 2, 3, 5, 6
Mason et al. [18]	2017	United States	Retrospective, single-centre PSM, IPTW	2003–2013	OPN, LPN, RAPN	410	1598	2
Neves et al. [19]	2023	United Kingdom	Prospective, single-centre Feasibility cohort-embedded RCT	2019–2021	RAPN	25	25	1, 2
Rembeyo et al. [20]	2019	France	Retrospective, single-centre IPTW	2010–2016	RAPN	55	36	1, 2, 3, 5, 6
Uemura et al. [21]	2021	Japan	Retrospective, single-centre	2016–2019	RAPN	48	78	1, 2, 3, 4, 5, 6
Yanagisawa et al. [22]	2020	Japan	Retrospective, multi-centre PSM	2011–2019	LPN	133	241	1, 2, 3, 4, 6

1: Complication rates; 2: renal function; 3: recurrence-free survival; 4: metastasis-free survival; 5: cancer-specific survival; 6: overall survival. Abbreviations: IPTW = inverse probability of treatment weighting; LPN = laparoscopic partial nephrectomy; No. = number; OPN = open partial nephrectomy; PCA = percutaneous cryoablation; PN = partial nephrectomy; PSM = propensity score matching, RAPN = robotic-assisted partial nephrectomy; RCT = randomised controlled trial.

**Table 2 cancers-16-01175-t002:** Clinicopathological characteristics of patients included in selected articles.

Author [Ref.]	Age (Years) PCA/PN	Male Gender, % PCA/PN	CCI Score PCA/PN	Solitary Kidney, % PCA/PN	Preoperative eGFR (mL/min/1.73 m^2^) PCA/PN	Biopsy before PCA, %	Clinical Stage, % PCA/PN	Tumour Size (cm) PCA/PN	RENAL Score PCA/PN	RCC Tumour, *n* (%) PCA/PN	Tumour Grade, % ^#^ PCA/PN	Residual Unablated Tumour after PCA, %	Positive Margin after PN, %	Follow-Up (Oncological Outcomes) (Months) PCA/PN
Aikawa et al. [9]	≤80: 85.5%/85.4% *	71.5/72.3 *	>3: 16.5%/15.8% *	14.6/13.9 *	≤30: 5.1%/6.5% *	62	cT1b: 100/100	4.5/4.5 ^b,^*	>9: 14.9%/14.8% *	14 (48)/90 (100)	NR	6.9	5.6	43/35.5 ^a^
Andrews et al. (*cT1a cohort*) [10]	72/62 ^a^	66/61	2/1 ^a^	NR	NR	93	cT1a: 100/100	2.8/2.4 ^a^	NR	108 (58)/835 (79)	G1: 24/21 G2: 48/68 G3: 8/10 G4: 0/0.4	NR	NR	75.6/112.8 ^a^
Andrews et al. (*cT1b cohort*) [10]	77/61 ^a^	75/68	2/1 ^a^	NR	NR	98	cT1b: 100/100	4.8/5 ^a^	NR	35 (67)/272 (84)	G1: 23/10 G2: 46/71 G3: 6/18 G4: 3/0	NR	NR	72/104.4 ^a^
Bhindi et al. [11]	65/63 ^a,^*	75/73 *	2/2 ^a,^*	100/100	56/56 ^a,^*	NR	cT1a: 60/57 * cT1b: 40/43 *	3.5/3.7 ^a,^*	7/8 ^a,^*	78/74 *	NR	NR	NR	Whole cohort: 47 ^a^
Bianchi et al. [12]	71/63 ^a^	71.1/65.9	NA	0/1.5	71/85 ^a^	100	cT1a: 91.6/80.8 cT1b: 8.4/19.2	2.2/3 ^a^	NR	65 (78.3)/606 (73.9)	NR	NR	NR	63/63 ^a^
Chan et al. (*cT1a cohort*) [13]	72/59 ^a^	58.3/65.8	3/2 ^a^	0/0	77.8/91.3 ^a^	100	cT1a: 100/100	2.9/2.5 ^a^	5/6 ^a^	72 (100)/79 (100)	G1: 23.6/6.3 G2: 48.6/34.2 G3: 8.3/48.1 G4: 1.4/5.1	NR	NR	75.6/72 ^a^
Chan et al. (*cT1b cohort*) [13]	77/57 ^a^	29.1/42.9	4/3 ^a^	0/0	57.6/84.8 ^a^	100	cT1b: 100/100	4.5/4.45 ^a^	9/7 ^a^	31 (100)/14 (100)	G1: 12.9/14.3 G2: 61.3/14.3 G3: 16.1/57.1 G4: 3.2/0	NR	NR	72.5/67.9 ^a^
Duus et al. [14]	68.5/57.5 ^b^	71.1/72.2	3/2 ^a^	7.8/0	71/91 ^a^	100	cT1a: 87.2/66.7 cT1b:12.8/33.3	3/3.6 ^a^	8/7 ^a^	39 (100)/56 (98.2)	NR	NR	NR	NA
Fraisse et al. [15]	69.9/59.9 ^b,^*	67.8/72.3 *	NR	NR	NR	100	cT1a: 96/96 * cT1b: 4/4 *	2.6/2.8 ^b,^*	>9: 5.7%/5.7% *	177 (100)/177 (100) *	NR	8.5 *	5.6 *	62.6/39 ^a^
Junker et al. [16]	69.4/63.6 ^a^	71/76	3/2 ^a^	0/0	NR	100	cT1a: 89/63 cT1b: 11/37	3.1/3.7 ^b^	8/7 ^a^	104 (100)/86 (100)	NR	NR	NR	NA
Kawaguchi et al. [17]	78/75 ^a^	71.4/68	NR	16.3/2	65.7/65 ^b^	100	cT1a: 93.9/84 cT1b: 4/16	2.4/2.7 ^b^	>9: 0%/2%	40 (81.6)/44 (72)	NR	NR	4	20.1/24.3 ^b^
Mason et al. [18]	67.7/66.7 *	67/68 *	NR	10/13 *	65.5/66.4 *	NR	NR	3.2/3.1 *	NR	NR	NR	NR	NR	NA
Neves et al. [19]	58.8/57.2	68/56	>3: 0/0	NR	84.7/83.7	NR	cT1a: 100/100	2.9/2/7	>9: 4%/8%	NR	NR	NR	NR	NA
Rembeyo et al. [20]	72/60 ^b^	67.3/77.8	NA	29.1/0	73/85	100	cT1b: 100/100	4.6/4.5	> 9: 34.6%/0%	44 (80)/32 (88.9)	G1: 6.8/0 G2: 52.3/53.1 G3: 15.9/28.1 G4: 2.3/9.4	NR	NR	19.9/23.7 ^a^
Uemura et al. [21]	78/61 ^a^	85.4/81	NR	4.2/0	53.6/73.2 ^a^	100	cT1a: 95.8/93.6 cT1b: 4.2/6.4	2.6/1.9 ^a^	>9: 10.8%/3.8%	48 (100)/75 (96.2)	PN group: G1: 37.2 G2: 51.3 G3:7.7	NR	1.3	12/18.5 ^a^
Yanagisawa et al. [22]	68.5/69.5 ^a,^*	76/81 *	1/1 ^a,^*	NR	62.5/63.2 ^b,^*	78	cT1a: 78/77 * cT1b: 12/13 *	2.76/2.88 ^b,^*	6/6 ^a,^*	65 (72.2)/90 (100) *	G1: 43/36 * G2: 49/60 * G3: 7.7/4.4 *	NR	NR	26.5/18 ^a^

^#^—proportion of histologically confirmed RCC tumours; ^a^—median; ^b^—mean; * values after matching. Abbreviations: CCI = Charlson Comorbidity Index; eGFR = estimated glomerular filtration rate; G = grade; NA = not applicable; NR = not reported; PCA = percutaneous cryoablation; PN = partial nephrectomy; RCC = renal cell carcinoma.

## Data Availability

No new data were created as part of this systematic review.

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
