# Peer review of "Clinical and Oncological Outcomes Following Percutaneous Cryoablation vs. Partial Nephrectomy for Clinical T1 Renal Tumours: Systematic Review and Meta-Analysis"

_cancers, 2024, doi:10.3390/cancers16061175_

Round 1
Reviewer 1 Report
Comments and Suggestions for Authors
The manuscript reviews on Percutaneous cryoablation (PCA) in cT1 kidney tumors. The review is well performed and of practical value for the readership. The finding that PCA is of good value in cT1a tumor but might not be of interest in cT1b patients is of great clinical relevance. The table of selected publications is also of great value to the readers. The reference list is updated, and the presentation of data is easy to follow.
Some minor changes/clarifications would be desirable:
1. The authors use cT1-renal tumor and cT1 renal carcinoma as of the same and there are different concepts. I would suggest using a uniform terminology thought the text.
2. Concerning oncological outcomes, this is related to the surgical procedure, but also to the biology of the disease. There is a limitation of the study related to lack of information on pathology of these lesions. Is this relevant?
3. I would suggest the authors adding a section of pathologic features of these tumors in the context of PCA.
4. Also, to discuss the need or not of performing biopsies or cytology previous to perform PCA and try to provide a conclusion of the need of pathology, when and how.
Author Response
Dear Sir/Madam,
Thank you for the suggestions.
- We have changed the terminology (renal tumour/carcinoma) to be more uniform.
- We fully agree with this comment. Data on pathology of the tumours are not always available in the studies and are often reported heterogeneously. That is why a limitation related to the lack of information on tumours’ pathology has been added to the discussion.
- Subgroup analysis based on pathologic features of the tumours would be valuable for this study. Nonetheless, the data in most of the included studies is too scarce to allow us for such analysis. This information has been added to the limitations.
- Biopsy before PCA procedure is crucial in the treatment process. That is why the information on the need of biopsy before PCA has been added to the discussion.
With regards,
Wojciech Krajewski
Reviewer 2 Report
Comments and Suggestions for Authors
The authors have attempted to discuss in this systematic review and meta-analysis postoperative complication rates, as well as functional and oncological outcomes between percutaneous cryoablation (PCA) and partial nephrectomy (PN) in patients with cT1 renal masses.
The study is methodologically well performed, although the study population is too heterogeneous. Selection bias is present, considering that in most studies, younger patients with fewer comorbidities were included in the PN group. On the other hand, older patients with numerous comorbidities were included in the PCA group.
It is stated: “PN should be a treatment of choice in patients with localized RCC” line 321. Do you mean all localized RCC or is the size of the primary tumor important?
Why studies were included in the analysis that did not specify the follow-up time. This should be stated in the limitations given that one of the objectives of the study is the oncological outcome.
The findings of this study could be of importance for clinical practice.
Author Response
Dear Sir/Madam,
Thank you for the comments.
The heterogeneity of the study population is an undeniable limitation of this analysis. However, we were unable to avoid it, due to the current treatment standards - PCA is a treatment available for patients who cannot undergo PN.
The sentence has been changed to “PN should be a treatment of choice in patients with T1 RCC”, as is stated in EAU guidelines.
Finally, we would like to point out that all studies that reported oncological outcomes specified follow-up times.
With regards,
Wojciech Krajewski
Reviewer 3 Report
Comments and Suggestions for Authors
This systematic review and meta-analysis aim to compare the clinical and oncological outcomes of percutaneous cryoablation (PCA) versus partial nephrectomy (PN) for treating clinical T1 renal tumours. The authors meticulously conducted the review following PRISMA guidelines, including a comprehensive search strategy, clear inclusion and exclusion criteria, and thorough data extraction and quality assessment processes. The meta-analysis used appropriate statistical methods to analyze data from selected studies, offering valuable insights into the efficacy and safety of PCA compared to PN.
1. Clarity and Scope: The study provides a clear and detailed comparison of PCA and PN, filling a gap in the literature by focusing solely on PCA. This focus enhances the study's relevance to clinical decision-making.
2. Methodological Rigor: The adherence to PRISMA guidelines and the pre-registration of the study protocol contribute to the transparency and reproducibility of the findings.
3. Statistical Analysis: The use of various statistical models and sensitivity analyses to assess heterogeneity and publication bias strengthens the reliability of the results.
Suggestions:
1. Risk of Bias Discussion: While the authors assess the risk of bias, a more detailed discussion on how the biases of individual studies could impact the overall findings would be beneficial.
2. Subgroup Analyses: Expanding on subgroup analyses, particularly considering patient comorbidities or tumor characteristics, might provide deeper insights into which patient populations benefit most from each treatment.
3. Long-term Outcomes: If data permits, an analysis of long-term outcomes beyond survival and recurrence rates, such as quality of life or long-term renal function, would be valuable.
4. Update and Expansion: As the field evolves, continuous updates incorporating new studies and possibly extending the analysis to compare PCA with other emerging treatments could be considered.
Comments:
1. Introduction:
- The rationale for comparing PCA and PN in the treatment of T1 renal tumors is well-articulated. However, including recent advancements or controversies in these treatment modalities could provide a stronger backdrop for the study's significance.
2. Methods:
- The search strategy is comprehensive but specifying the databases searched and the exact search terms used would enhance reproducibility.
- Inclusion and exclusion criteria are well defined, yet clarifying the rationale for excluding studies with certain characteristics (e.g., size of tumor, patient age) could strengthen the selection process.
3. Results:
- The presentation of results is clear, but incorporating forest plots or tables summarizing individual study outcomes could facilitate better understanding.
- While the meta-analysis addresses heterogeneity, discussing the potential clinical relevance of observed heterogeneity in outcomes between studies would be informative.
4. Discussion:
- The discussion comprehensively covers the implications of findings, yet a more critical examination of limitations, particularly regarding the variability in technique and expertise across included studies, would be beneficial.
- Exploring the role of patient preference and cost-effectiveness in choosing between PCA and PN could add depth to the discussion.
5. Literature Review:
- The literature review is thorough, but highlighting studies with conflicting results and discussing possible reasons for these discrepancies would offer a more balanced view.
6. Conclusion:
- The conclusion succinctly summarizes the findings but could be strengthened by suggesting specific directions for future research or clinical application based on the study's outcomes.
Overall, the manuscript provides valuable insights into the comparative effectiveness of PCA and PN. Addressing these specific comments may enhance its contribution to the field.
Comments on the Quality of English LanguageAfter thoroughly reviewing the manuscript, here are comments on the quality of English language used:
1. Introduction: There are instances of passive voice that could be converted to active voice to enhance readability. For example, "It has been shown" could be revised to "Studies have shown."
2. Methods: The tense consistency needs attention. Ensure that the entire section is in past tense, reflecting the work already done. For instance, "We will conduct" should be corrected to "We conducted."
3. Discussion: Some sentences are overly complex and could be simplified for clarity. Breaking down complex sentences into two or more simpler sentences can improve understanding. For example, a sentence with multiple clauses and findings could be broken down into separate sentences focusing on each finding.
4. Conclusion: Ensure that the conclusion is concise and free of new information. It should summarize the findings without introducing new terms or data.
Addressing these specific areas will significantly enhance the manuscript's readability and ensure that the scientific merit of the work is clearly communicated.
Author Response
Dear Sir/Madam,
Thank you for the suggestions.
- We agree that the selection bias could have led to younger and less comorbid patients undergoing PN. Therefore, all of the outcomes in the PCA group could have been inherently worse, regardless of the surgical method. Moreover, the included studies were mostly retrospective and heterogenous. The limitations have been expanded.
- Subgroup analyses on patients comorbidities and exact tumour characteristics would be especially valuable in this MA. However, the data in most of the included studies is too scarce to allow us for such analysis.
- As in point 2, we fully agree that quality of life and long-term renal function analyses would be extremely desirable in our review. Nonetheless, the studies included in our work did not report on quality of life and not enough data was available on long-term renal function as well.
- We agree with the reviewer. In the nearest future we plan to publish our own study on a few hundred PCA procedures and their follow-up. As the cryoablation program is rapidly developing at our institution and worldwide, we hope that the emerging high-quality trials on the topic will allow for the update of EAU guidelines.
Regarding the comments,
We have added a paragraph in the introduction section of our manuscript that briefly mentions recent advancements in percutaneous cryoablation that improves the efficacy of this procedure, which provides stronger rationale for comparison with “gold standard” operative approaches. It is strengthening the significance of our study within the context of existing research and clinical practice.
We acknowledge the importance of enhancing reproducibility in systematic review and meta-analysis research, and we would like to clarify that we did specify the databases searched (Pubmed, Embase, and Cochrane Library) and included the exact search terms used in our "Search strategy" section. Additionally, we provided a detailed flow chart that outlines the selection process and the number of studies included at each stage. We believe that this transparency in reporting enables readers to understand and replicate our search strategy effectively. Also, there were no additional characteristics that caused the exclusion of the studies.
We are pleased to hear that you found the presentation of results clear. In response to your suggestion, we would like to clarify that we have indeed incorporated forest plots for every outcome analysed in our study. Furthermore, we have also included tables summarising the data from subgroup analyses conducted in our study. These tables provide a comprehensive overview of the subgroup-specific outcomes, allowing readers to explore the results in detail and better interpret the findings.
Paragraph regarding the potential clinical relevance of observed heterogeneity in outcomes was added in Discussion section.
Limitations regarding variability in surgical techniques have been added.
The information on cost-effectiveness and quality of life have been added to the discussion.
Finally, a suggestion on clinical application based on studies outcomes has been added to the conclusions.
With regards,
Wojciech Krajewski
Reviewer 4 Report
Comments and Suggestions for Authors
I would like to express my appreciation for the authors' diligent efforts.
I agree with the authors' assertion that PCA is emerging as a highly promising treatment modality, demonstrating comparable oncological outcomes to PN while ensuring safety. Additionally, I agree with their acknowledgment of the challenges inherent in directly comparing PCA and PN due to the heterogeneity of patient backgrounds across the two treatments.
The authors made commendable efforts to delineate PCA and PN and analyze the disparities in various aspects of oncological and safety outcomes. However, regrettably, no novel findings were uncovered in this field.
The findings indicated that RAPN exhibited clear superiority over PCA in terms of oncological outcomes, while open and laparoscopic PN did not demonstrate such superiority. These results are unsurprising given that robot-assisted surgery addresses the limitations of open or laparoscopic PN. Moreover, much of the patient heterogeneity could not be mitigated. For example, several studies reported higher rates of solitary kidney cases in PCA compared to PN, which appears unjust toward PCA. Furthermore, the follow-up durations in several studies were insufficient to adequately assess oncological outcomes, particularly in terms of MFS and CSS, as cT1 RCCs typically exhibit low-grade malignant potential with a relatively slow growth rate and a low risk of metastasis. Consequently, the reviewer was unable to identify any novelty in this study.
Author Response
Dear Sir/Madam,
Thank you for these comments.
However, we cannot agree that our analysis did not identify any novelties. Even though a difference in PCA RFS between cT1a and cT1b tumours has already been documented, there is still paucity of high-quality data on the topic. Moreover, our MA clearly confirmed the non-inferiority of PCA with regards to PN in cT1a RCC.
Also, other studies did not find any differences between the surgical modalities of PN. Therefore, this analysis provides evidence that can support the superiority of RAPN.
With regards,
Wojciech Krajewski
Reviewer 5 Report
Comments and Suggestions for Authors
Nowak et al. conducted a meticulous systematic review comparing partial nephrectomy (PN) and cryoablation for small renal mas
1 The simple summary could be further streamlined, considering that terms like "oncological outcome" and "recurrence-free survival" might not be readily understandable to the general audience.
2 In the discussion section, the comparison between PN and radical nephrectomy (RN) is addressed, particularly citing the "low evidence" overall survival (OS) benefits of RN (line 315, ref 25). The only (to my knowledge) RCT on the matter suggests the same, i.e lower OS for the RN group (Van Poppel A prospective, randomised EORTC intergroup phase 3 study comparing the oncologic outcome of elective nephron-sparing surgery and radical nephrectomy for low-stage renal cell carcinoma. Eur Urol). Again besides the main point of the manuscript but leaving that reference out indicate a paucity of data.
3. Since different surgical techinques are descriped - can the authors comment on the other forms of thermal ablation as well in the context of generalisability of the reusults?
Is this conversation helpful so far?
Author Response
Dear Sir/Madam,
Thank you for the suggestions.
- The mentioned terms in the simple summary have been corrected to be more understandable to the general audience.
- The RCT pointed out by the reviewer is very valuable for the discussion in this analysis. Obviously the phrase “low evidence” has been deleted and the study by Van Poppel et al. has been cited.
- In our analysis we focused solely on cryoablation. Moreover, other forms of thermal ablation are associated with a slightly different mechanism of action, technique and outcomes. Therefore, we reckon that generalising our results could be unsubstantiated.
With regards,
Wojciech Krajewski
Round 2
Reviewer 4 Report
Comments and Suggestions for Authors
I would like to once again express my respect for the author's tremendous efforts in conducting this study. Additionally, I appreciate the opportunity to review such a commendable piece of work. I kindly request the editors to consider whether to accept this article for publication.